# Two Old Wild-Type Strains of *Drosophila melanogaster* Can Serve as an Animal Model of Faster and Slower Aging Processes

**DOI:** 10.3390/insects15050329

**Published:** 2024-05-03

**Authors:** Lyudmila P. Zakharenko, Margarita A. Bobrovskikh, Nataly E. Gruntenko, Dmitrii V. Petrovskii, Evgeniy G. Verevkin, Arcady A. Putilov

**Affiliations:** Department of Insect Genetics, Institute of Cytology and Genetics of the Siberian Branch, The Russian Academy of Sciences, Novosibirsk 630090, Russia; zakharlp@bionet.nsc.ru (L.P.Z.); eremina@bionet.nsc.ru (M.A.B.); nataly@bionet.nsc.ru (N.E.G.); dm_petr@bionet.nsc.ru (D.V.P.); ewer@mail.ru (E.G.V.)

**Keywords:** fruit fly, model organism, circadian rhythm, sleep, locomotor activity, longevity, fecundity, glucose, trehalose

## Abstract

**Simple Summary:**

The life expectancy of *Drosophila melanogaster* evolves in the laboratory in a downward direction over time. The comparison of two old laboratory-adapted strains, Canton-S and Harwich, suggested that they may be used as a multicellular eukaryote model for studying the effects of individual difference in aging processes on the physiological responses to various environmental factors, including low and high temperature, carbohydrate-heavy food, and caffeinated food in the presence of light-dark cyclicity and constant darkness. We found that the accelerated aging in Harwich was associated with a shorter lifespan, lower fecundity, faster rate of development, smaller body weight, lower concentrations of two main insect sugars, reduced locomotor activity, and larger amount of sleep. Such findings imply that, in the experimental studies of the aging accelerated effects of various adverse factors, the results obtained with only one of the laboratory-adapted strains cannot be generalized to other strains. The documented individual and strain-associated differences in the responses of their daily rhythms to aging and age-accelerated factors allow the recommendation to use at least two distinct strains characterized by a relatively fast and relatively slow aging processes, for the experimental elaboration of relationships between genes, environment, behavior, physiology, and health.

**Abstract:**

Background: *Drosophila melanogaster* provides a powerful platform to study the physiology and genetics of aging, i.e., the mechanisms underpinnings healthy aging, age-associated disorders, and acceleration of the aging process under adverse environmental conditions. Here, we tested the responses of daily rhythms to age-accelerated factors in two wild-type laboratory-adapted strains, Canton-S and Harwich. Methods: On the example of the 24 h patterns of locomotor activity and sleep, we documented the responses of these two strains to such factors as aging, high temperature, carbohydrate diet, and diet with different doses of caffeine-benzoate sodium. Results: The strains demonstrated differential responses to these factors. Moreover, compared to Canton-S, Harwich showed a reduced locomotor activity, larger amount of sleep, faster rate of development, smaller body weight, lower concentrations of main sugars, lower fecundity, and shorter lifespan. Conclusions: It might be recommended to use at least two strains, one with a relatively fast and another with a relatively slow aging process, for the experimental elaboration of relationships between genes, environment, behavior, physiology, and health.

## 1. Introduction

*Drosophila melanogaster* is the most known vinegar fly species or, less correctly, fruit fly species. This is a human commensal of eastern sub-Saharan African origin [1,2]. Since the beginning of the last century, it has served as a model organism for studying a diverse range of biological and medical phenomena, including aging and development, vulnerability to various diseases, locomotion and other behaviors, learning and cognitive skills, evolution through natural and artificial selection, genetics and inheritance, etc. *Drosophila melanogaster* has become one of the powerful multicellular eukaryote models for studying these biomedical phenomena due to its rapid life cycle, ease of maintenance in the laboratory, well-understood genetics, only having four pairs of chromosomes, sequenced genome, cosmopolitan distribution, and natural ecology that provides substantial genetic variation across significant environmental heterogeneity [3,4,5].

Although humans and flies do not look very similar, most of the fundamental biological mechanisms and pathways that control an organism’s aging, development, metabolism, survival, reproduction, and behavior are conserved between these two species across their evolution. Therefore, the genes affecting aging in Drosophila often have human orthologs and will elucidate corresponding mechanisms in our species [6,7]. Given the complexity and limitations of human studies, *Drosophila melanogaster* can be regarded as a very important model organism for dissecting relationships between genes, environment, behavior, physiology, and health [8].

In particular, *Drosophila melanogaster* provides a powerful and rapid platform to study the physiology and genetics of aging, i.e., the mechanisms underpinnings healthy aging, age-associated health disturbances, and acceleration of the aging process due to the influence of adverse environmental factors [5,9,10].

It seems that the life expectancy of *Drosophila melanogaster* evolves in the laboratory in a downward direction over time [11]. The comparison of wild-caught flies with flies from the same population that had been maintained in the laboratory for several generations suggested that laboratory culture leads to an increase in early fecundity and a decline in longevity. Most likely, such changes are observed (1) due to the accumulation of late-acting deleterious mutations and (2) because laboratory culture selects for increased early fecundity, which increases the costs of reproduction, thereby increasing mortality [12].

For example, since as early as in 1935, the stock of one of the most used wild-type strains in *Drosophila melanogaster* studies, Canton Special (Canton-S), has been established by C. B. Bridges [13,14], and, as it has been found 65 years later, this laboratory-adapted strain lives for a significantly shorter time than wild-caught strains [15]. Canton-S is one of the two reference strains used to test dysgenic potential (see Appendix A for more details). Another strain, called Harwich, was collected in the wild in 1967 by M. L. Tracey, Jr. and, one decade later, its white-eyed mutant was isolated and standardized by M. G. Kidwell [16].

Since Harwich had evolved to become reproductively isolated from Canton-S in the condition of high temperatures, its study is of special interest to comparing life-history traits in these two strains. Particularly, it remains to be elucidated whether, similarly to the fecundity and lifespan of Canton-S, these life-history traits of Harwich have also evolved in the laboratory in the direction of earlier fecundity and shorter lifespan, and if yes, which of the two strains has advanced further in this direction.

Consequently, one of the major purposes of the present study was to test whether Canton-S and Harwich differ in such life-history traits as longevity, fecundity, rate of development, metabolism, level of activity, and amount of sleep. Although the two latter traits are not usually listed among life-history traits, their importance in this respect can be exemplified by the fact that *Drosophila melanogaster* needs to be more active and sleep less to adapt to either a longer or harder life [17].

Therefore, we also purposed this study on testing whether the difference between Canton-S and Harwich can be associated with the differences in the speed of age-induced changes in their 24 h patterns of locomotor activity and sleep as well as with their differential responses to several adverse factors accelerating the aging process.

These differential responses were examined by comparing these two strains on the modifications of their 24 h patterns of locomotor activity and sleep in response to exposure to such different factors as high temperature, permanent darkness, and caffeinated and low protein/high carbohydrate food. If the difference in the speed of aging process and their differential responses are experimentally supported, such results imply that findings obtained with only one of these laboratory-adapted strains cannot be generalized to other strains, and, at least two distinct (in this respect) strains should be used for the experimental elaboration of relationships between genes, environment, behavior, physiology, and health.

## 2. Results

Appendix A summarize the results of analyses of experimental data on longevity (Appendix A), fecundity (Appendix A), body weight, concentrations of sugars (Appendix A), the responses of daily rhythms of locomotor activity and sleep to changes in age, temperature, light-darkness regime, food quality, and concentrations of caffeine-benzoate sodium (Appendix A), and the rates of acceleration of age-associated changes in the 24 h pattern of locomotor activity and sleep (Appendix A).

### 2.1. Longevity

As was suggested by analysis of survival curves (Figure 1 and Appendix A) using the Log Rank (Mantel–Cox) test of equality of survival distributions (Appendix A), under high temperature, longevity is more profoundly reduced in Harwich than in Canton S. The comparison of survival curves of parental strains and their F1 and F70 hybrids revealed a significantly higher survival rate for infertile and fertile female hybrids compared to parental strains (Appendix A). Moreover, the results confirmed that longevity is more profoundly reduced in Harwich than in Canton S (Appendix A).

Figure 1 also illustrates the result of adding different doses of caffeine-benzoate sodium solution to the standard food of flies from the parent strains. Whereas a small dose of caffeine-benzoate sodium (×1) does not significantly change the survival curve of Canton S, this dose causes further significant reduction of lifespan in Harwich (Figure 1 and Appendix A). Only a tenfold-larger dose (×10) reduced the survival rate in both of the two strains (Figure 1 and Appendix A), thus yielding the negative health effect of adding such a caffeine-benzoate sodium solution to the standard food of flies.

### 2.2. Fecundity

The results of rANOVA of two experiments on fecundity yielded significant interaction of the factor “Strain” with the day of collection of hatching imagoes (Appendix A). Figure 2 illustrates this result, indicating a slower development of Canton-S compared to Harwich. Namely, after initial collection of offspring in the first day, the number of collected Canton-S did not decline the next day, while this number always declined in Harwich (Figure 2). As for the total number of offspring, a significant main effect of the factor “Strain” pointed at a higher fecundity of Canton-S than Harwich (Appendix A and Figure 2 and Appendix A). The difference between these strains was most notable in the case of inclusion of only one female parent in reproduction (Appendix A).

### 2.3. Fly Weight and Concentrations of Two Main Insect Sugars

The differences between the two strains in longevity, fecundity, and rate of development are expected to be associated with their differences in several important morphological and metabolic indicators of fitness, such as fly weight and concentrations of the two main insect sugars, trehalose and glucose. Indeed, a significantly higher fly weight of Canton-S than Harwich was yielded by two-way ANOVA (Appendix A).

Sugars are used primarily by an organism for the metabolic production of ATP energy and carbon sources. Despite the close homology of insects and vertebrates in their sugar metabolism, there is one profound difference between them: insects have trehalose (or α1-D-glucopyranosyl-α1-D-glucopyranoside), which is a much more important source of their energy than glucose. Therefore, two major carbohydrates, trehalose and glucose, serve as the circulating energy sources in *Drosophila* hemolymph [18], with higher levels of trehalose in the hemolymph being essential to meet the energy demands of flight muscles [19].

Results of three-way rANOVA of these two sugars suggested that the higher longevity and fecundity and slower development in Canton-S can be linked to higher circulating glucose and trehalose levels.

As illustrated in Figure 3, the four groups, male and female Canton-S and male and female Harwich, do not overlap in the two-dimensional space of the concentration of glucose and trehalose. Female Canton-S are the champions on the concentration of glucose and their concentration of trehalose resembles that of the males of the two strains, while, in contrast, the concentration of trehalose in female Harwich is the lowest among the four groups and their concentration of glucose resembles that of males of the two strains, i.e., it is lower than in female Canton-S (Figure 3).

### 2.4. Daily Averaged Levels of Locomotor Activity and Sleep

Given the difference between Canton-S and Harwich in concentrations of sugars, it is expected to find a higher level of male locomotor activity in Canton-S than Harwich in the results of the 6 rANOVAs reported in Appendix A. Irrespective of the influence of other experimental factors, the main effect of the factor “Strain” on locomotor activity was always found to be significant at *p* < 0.001 (Appendix A and Figure 4). Accordingly, the main effect of this factor on sleep was also significant at *p* < 0.001, with a smaller amount of sleep in Canton-S than in Harwich (Appendix A).

The interaction of the factor “Strain” with experimental interventions was significant for manipulations with food. The low protein/high carbohydrate diet reduced locomotor activity and increased amount of sleep more remarkably in Canton-S than in Harwich (Figure 4D and Appendix A). Figure 4A,C and Appendix A illustrate the significant decline of locomotor activity and significant rise of amount of sleep with age, but the interaction between the factors “Strain” and “Age” was mostly non-significant (Appendix A).

### 2.5. The 24 h Patterns of Sleep and Locomotor Activity

In contrast with the interactions obtained for daily averaged levels of locomotor activity and sleep, the 24 h patterns in the two strains demonstrated clearly differential responses to aging and adding various adverse factors. Such differential responses are indicated by the significant interaction of “Time” (48 time points on 24 h interval) and “Strain” with such factors as “Age”, “Temperature”, “Dose”, and “Food”, and with the combinations of these factors (Appendix A). First of all, these differential responses can be exemplified by the age-associated changes in the patterns illustrated in Figure 5 and Appendix A. Under high temperature, the morning peaks of locomotor activity (Appendix A) and wakefulness (Figure 5) became smaller in both strains, but these peaks almost fully disappeared already at a young age of flies only in Harwich, while it remained clearly noticeable in Canton-S. With advancing age, the evening peaks of locomotor activity and wakefulness shifted to earlier hours, but this trend was clearly seen much earlier in Harwich than in Canton-S. The intervals of low locomotor activity and high amount of sleep rapidly increased with age in Harwich. In contrast, the aging of Canton-S was associated with insomnia-like symptoms. Flies of this strain became almost fully arhythmic in old age, while old Harwich flies demonstrated hypersomnia-like symptoms with the shift of peaks of locomotor activity and wakefulness to the earlier hours (Figure 5 and Appendix A, respectively).

High temperature caused the delay of evening peaks of locomotor activity and wakefulness combined with either insomnia-like symptoms in Canton-S or hypersomnia-like symptoms in Harwich. “Postlaunch deep” became prominent in both strains, but the amount of sleep during the day increased in Harwich, while it did not change much in Canton-S (Figure 5B,C and Appendix A). The morning peaks disappeared after adding 10 doses of caffeine-benzoate sodium solution to the standard diet (Figure 5C and Appendix A).

The differential responses of the 24 h patterns are also found on the example of response to high temperature illustrated in Figure 6 and Appendix A. This response is indicated by significant interaction of “Time” and “Strain” with the factor “Temperature” (Appendix A).

The differential responses of the 24 h patterns observed in the two strains are also illustrated in Figure 7, Figure 8, Appendix A.

Figure 7 and Appendix A show these patterns in young and old flies from the two strains fed with standard food either with or without adding caffeine-benzoate sodium in different doses. None of the interactions of “Time” and “Strain” with “Age” and/or “Dose” were found to be significant (Appendix A). Delays of evening peaks and insomnia-like symptoms were clearly detected exclusively in Canton-S, while Harwich responded to aging and adverse food by the shift of its only peak from evening to daytime and hypersomnia-like symptoms (Figure 7, Figure 8, Appendix A). The remaining age-associated changes were found in flies of the two strains under the influence of the adverse factors of a low protein/high carbohydrate diet, either alone or in combination with caffeine-benzoate sodium (Figure 8 and Appendix A). Any possible interactions of “Time” and “Strain” with “Food” and/or “Dose” reached the statistically significant level, including the interaction between all these factors (Appendix A).

Finally, Figure 9 and Appendix A illustrate the difference between the 24 h patterns observed in constant darkness and under the typical laboratory light-dark cycle (12 h of light and 12 h of darkness). The morning peaks became clearly seen in both strains under the periodic environmental cycles, but their differential response to adding caffeine-benzoate sodium to the standard diet persisted under any light condition (Figure 9 and Appendix A). Therefore, the triple interactions of “Time” and “Strain” with either the factor “Dose” or the factor “Cycle” were significant (Appendix A).

### 2.6. Principal Component Scoring of the 24 h Patterns of Sleep and Locomotor Activity

Appendix A and Figure 10 and Appendix A illustrate the results of quantitative comparison of strains and their crosses on the rate of age-induced changes in their 24 h patterns.

It turns out that the individual variation in scores on the 1st principal component of variation in locomotor activity and sleep under high air temperature reflects the age-induced transition from an increased to a decreased level of locomotor activity and from a decreased to an increased amount of sleep, respectively, i.e., from “young age” to “old age” levels. Therefore, positive and negative scores on this component separate the flies characterized by a higher and lower locomotor activity and a lower and higher amount of sleep, respectively (Figure 10A–C and Appendix A). Individual variation in scores on the 2nd principal component reflects the age-induced modification of the daily patterns of locomotor activity and sleep. With advancing age, they transform from an almost bimodal daily rhythm with a clear evening peak of locomotor activity and a clear evening trough of sleep (i.e., “young age” daily patterns) to the definitely unimodal daily rhythm with the only peak and the only trough of locomotor activity and sleep, respectively, shifted to daytime hours (i.e., “old age” patterns). Therefore, negative and positive scores separate the flies having an almost bimodal rhythm with a clear evening peak and trough from the flies demonstrating the unimodal rhythm with a single peak and a single trough, respectively (Figure 10A–C and Appendix A). As expected, we found that the aging process is associated with a decrease of rate of flies with “young age” scores and an increase of rate of flies with “old age” scores (Figure 10D). At any age, the differences between flies from the parent strains in the rate of “old age” scores almost always reached a significant level with a higher rate of “old age” scores in Harwich than in Canton-S, while their crosses had intermediate scores (Appendix A and Figure 10D). This result suggests that the speed of acceleration of sleep aging in the crosses seems to be intermediate between a faster speed in Harwich and a slower speed in Canton-S.

## 3. Discussion

It was previously found that a wild-type laboratory-adapted strain such as Canton-S lives for a significantly shorter time than wild-caught strains [15]. Here, we showed that another laboratory-adapted strain, Harwich, lives for an even shorter time. Moreover, we demonstrated that it has lower fecundity, a faster rate of development, smaller body weight, lower concentrations of two main insect sugars, reduced locomotor activity, and a larger amount of sleep. Finally, we demonstrated, on the example of the 24 h patterns of locomotor activity and sleep, the differential responses of these two strains to various adverse aging-accelerated factors, such as high temperature, a low protein/high carbohydrate diet, and a diet with different doses of caffeine-benzoate sodium. Such findings imply that, in experimental studies of the aging-accelerated effects of these adverse factors, the results obtained for only one of the two strains cannot be generalized to other strains. Consequently, the present results allow the recommendation to use, at least two strains, one with a relatively fast and another with a relatively slow aging process, for the experimental elaboration of relationships between genes, environment, behavior, physiology, and health.

We expected that the adverse aging-accelerated factors would affect the daily averaged levels of locomotor activity in different directions, either increasing it (on the example of high temperature), or reducing it (on the example of low protein/high carbohydrate diet), or either reducing or increasing it (on the example of diet with different doses of caffeine-benzoate sodium). Of the particular results pointing at the difference in responses of the two strains, the most interesting finding was the result of the comparison of the two strains on the influence of the aging process on their 24 h patterns of sleep and locomotor activity when challenged by high temperature under constant darkness. The responses resemble the effects of age on the human sleep-wake cycle in several respects. First, the sleep of flies tends to peak only once, not twice, during the day, which makes this pattern similar to the monophasic human sleep. Second, when people and flies get older, this single peak tends to shift to earlier hours, and this tendency is seen much earlier in Harwich than Canton-S. Third, people of old age, like the flies of old age, suffer in the tested conditions from the symptoms of either insomnia or hypersomnia. Fourth, the flies from the oldest of the tested age groups tend to sleep either more or less than in younger ages (Harwich or Canton-S, respectively). This pattern of change in sleep in the later stages of life resembles the previously reported association of the increase of the risk of all-cause mortality in humans with a 3-year change either from a short to long sleep duration, or from a long to short sleep duration [20]. Thus, the two strains of *Drosophila melanogaster* can be used for the elaboration of individual differences in physiological and genetic mechanisms underlying such age-associated changes in the 24 h sleep pattern.

Hot ambient temperature belongs to one of the most powerful natural factors that cause profound disturbances in the human sleep-wake cycle [21,22]. Since the 24 h sleep pattern in *Drosophila* is very sensitive to heat, it is enticing to draw parallels to the sleep-wake behavior of humans exposed to thermal stress [23]. The experimental research indicates that the fly’s sleep-wake pattern might be reorganized by an increase of ambient temperature in a way that is very similar to the response of the human sleep-wake pattern to heat. For instance, nighttime sleep and daytime activity usually decrease, whereas daytime sleep and early night activity usually increase after exposure to high temperature [24,25,26,27]. Similar responses were confirmed in the present study that additionally showed that such particular modifications of sleep patterns can vary from one strain to another. Again, the amount of night sleep profoundly decreased in Canton-S, whereas the levels of daytime activity decreased to almost zero in Harwich.

The present results are in agreement with previously published reports indicating the adverse health effects of caffeine and sodium benzoate in *Drosophila melanogaster* [28,29,30]. One of the novel findings was a decrease rather than an increase of activity (and an increase rather than a decrease of amount of sleep) in response to such a caffeinated diet. Such a response differs from the well-known response of humans to caffeine. Since bitter taste perception contributes to the coffee drinking behavior of *Drosophila* [31,32], this activity reducing effect might be related to the modifying action of sodium benzoate on test perception. Thus, in contrast to aging and high-temperature responses, the effect of the examined caffeine-containing diet does not demonstrate any resemblance with the well-known effect of caffeine on humans. Nevertheless, the differential responses of the two strains were documented not only on the examples of aging and high-temperature response but also on the examples of the response to combinations of caffeine-containing and low protein/high diet carbohydrate diets.

The results of the comparison of the parent strains with their crosses revealed additive genetic effects on the age-induced change in the 24 h patterns of activity and sleep. Therefore, further research can be directed to artificial selection among hybrids to isolate the genes involved in the variability in speed of such changes.

Another direction of future research could be to identify the most powerful causes of acceleration of the aging process in Harwich compared to in Canton-S. Reactive oxygen species might be one of several such causes [33,34,35]. We previously studied the intracellular pattern of accumulation of reactive oxygen species in control and dysgenic germ tissue taken from the female hybrids and found a much faster accumulation of reactive oxygen species in dysgenic tissue [36]. Therefore, such future research can be aimed at the comparison of these two strains on their age-associated accumulation of reactive oxygen species.

## 4. Materials and Methods

Flies of the strains were received in 1996 from L.Z. Kaidanov (Canton-S) and in 1990 from the Obninsk Drosophila collection (Harwich). Starting from these dates, they were cultivated at the Institute of Cytology and Genetics (Novosibirsk). To test hybrids of the two strains, females of the Canton-S (♀C-S) strain were crossed with males of the Harwich (♂H) strain (♀C-S♂H), and, in turn, Harwich females were crossed with Canton-S males (♀H♂C-S). F1 hybrids were obtained by breeding 10 females and 10 males in each tube. To obtain F70 hybrids, ♀H♂C-S and ♀C-S♂H and further crosses were maintained at 25 °C (i.e., this temperature does not cause sterility in F1-F70 female progeny, see Appendix A).

The majority of methods applied for the collection and analysis of data in the present study were previously described in two of our previous publications [17,37].

### 4.1. Testing Longevity

In total, 1400 flies (700 per strain) were compared for longevity twice. To obtain survival curves, 20 flies were kept per one tube under a natural photoperiod and high temperature, 29 °C. The food vials were replaced, at least twice a week after removing and counting dead flies.

In the first experiment, males and females from the two strains (400 flies per strain) were compared (Appendix A).

In the second experiment (300 male flies per strain), different doses of caffeine-benzoate sodium (produced by Borimed, Borisov, Minsk region, Belarus) were added to standard food (Appendix A, three next lines, and Figure 1). The concentrations of caffeine used in the present experiments were 0.2 (×1), 0.5 (×2.5), and 2.0 mg/mL of food (×10.0).

To compare the survival curves of parental strains with the curves of their offspring (♀C-S♂H and ♀H♂C-S), 2000 flies were tested twice under high temperature using two generations of crosses (either F1 and F70). In each of the tubes, 20 male or female flies were kept at 29 °C, and the food vials were replaced twice a week after removing and counting dead flies (Appendix A).

### 4.2. Testing Fecundity and Fly Weight

Fecundity was tested in 358 groups of flies aged 5 days old. To test fecundity, male and female parents were kept together for one or five days at different temperatures. The number of hatching imagoes was counted twice (Appendix A and Figure 2 and Appendix A).

In the first experiment with two temperatures, each of the 248 mixed sex groups consisted of either one male and two females or two males and one female (Appendix A). Egg laying lasted five days. The 4th and 8th day of hatching were chosen for counting offspring (Appendix A).

In the second experiment with three temperatures, each of the 110 groups consisted of 5 pairs of flies (Figure 2). Eggs laying lasted one day. The 1st and 2nd day of hatching were chosen for counting offspring (Figure 2).

In order to determine the weight of 10 randomly chosen 5-day flies, we used weighing scales produced by the Ohaus Corp (Pine Brook, Parsippany-Troy Hills, NJ, USA). In total, the fly weight was measured in 9 groups of male flies and 9 groups of female flies from each strain (Appendix A). The weight of one fly was calculated by dividing the value obtained for the whole group of 10 flies by 10.

### 4.3. Measurement of Concentrations of Two Sugars

The concentrations of two main insect sugars, glucose and trehalose, were measured in 9 samples of each of the two sexes from each of the two strains (Figure 3 and Appendix A). To measure these concentrations (Appendix A), we used an assay that was originally developed by Musselman et al. [38] and slightly modified by Karpova et al. [39]. Carbohydrates were measured using a SmartSpec SmartSpec Plus (Bio-Rad, Philadelphia, PA, USA) at a wavelength of 340 nm. Trehalose was converted into glucose by adding trehalase (Sigma-Aldrich, St. Louis, MO, USA—T8778, 0.5 units/mL) with a further measurement of glucose titer using a Glucose (HK) Assay Kit (Sigma-Aldrich, St. Louis, MO, USA—GAHK20).

### 4.4. Testing Various Adverse Effects on Sleep and Locomotor Activity

The averaged 24 h patterns of locomotor activity and sleep were obtained for 2052 flies of the two strains. Prior to the recording of locomotor activity and sleep, the groups of 20–25 male flies from a strain were kept in standard vials under a standard temperature (25 °C) and photoperiod (light between 7:00 and 19:00). The conventional approach [40] was applied to acquire and analyze locomotor activity using a DAMS (Drosophila Activity Monitoring System; “Trikinetics”, Waltham, MA, USA) and the original software package (see the TriKinetics web site: www.trikinetics.com, accessed on 12 March 2024).

To record locomotor activity, each fly was individually placed in a glass locomotor-monitoring tube of the DAMS with three sets of infrared beams for activity detection. To record beam breaks with one-minute intervals, the monitor was connected to a computer. As is typical, locomotor activity was recorded in constant darkness for at least 5 days and initially expressed as the number of beam breaks in 1 min bins.

The initial counts of locomotor activity were additionally used to quantify sleep events. In accordance with the Donelson et al. [41] criterion, sleep was defined as 5 consecutive minutes of the absence of any locomotor activity. On the basis of this approach (www.trikinetics.com, accessed on 12 March 2024), our own excel software was developed and used for further conventional analysis of locomotor activity and sleep over longer intervals. One-minute data were averaged on the 30 min intervals of each record prior to applying statistical analysis (Appendix A) and prior to drawing illustrations of the group-averaged results (Figure 4, Figure 5, Figure 6, Figure 7, Figure 8, Figure 9 and Appendix A).

The two strains were compared at as many as four different ages (several days, one week, two-three weeks, and four-five weeks), under different regimes of illumination (the standard light-dark cycle and constant darkness), under low and high temperature (20 °C and 29 °C, respectively), on different diets of standard food (18 g of dry yeast, 50 g of corn grits, 20 g of sugar, 40 g of raisins, 5.6 g of agar, and 7 mL of nipagin at 10%) and low protein/high carbohydrate food (7 g of dry yeast, 50 g of sugar, 12 g of starch, 6.4 g of agar, and 4 mL of nipagin at 10%), and without and with adding up to two different doses of caffeine-benzoate sodium.

### 4.5. Methodology Developed for Comparison of Age-Induced Changes in the 24 h Patterns

The following additional approach was developed for quantitative comparison of strains and their crosses on the speed of age-induced changes in their 24 h patterns of locomotor activity and sleep. The individual sets of 48 30 min values from a 24 h interval (n = 428) were subjected to principal component analysis with extraction and scoring of the two largest principal components (PC1 and PC2). These two components explained 23% and 4% of the total individual variation in activity, respectively, and 18% and 6% of the total individual variation in sleep, respectively. Loadings of the 48 30 min intervals on PC1 and PC2 (Figure 10A) and the 24 h patterns of locomotor activity (Figure 10B and Appendix A) and sleep (Figure 10C and Appendix A) of flies with PC1 and PC2 scores ≤0 and >0 were found to be accurate indicators of the age-associated trends in the levels and daily patterns of activity and sleep. Therefore, the rates of flies with positive and negative PC1 and PC2 scores were calculated for the parent strains and their crosses (♀C-S♂H) of four ages (Figure 10D) and statistically compared by applying the Chi-square test (Appendix A).

## 5. Conclusions

The life expectancy of *Drosophila melanogaster* evolves in the laboratory in a downward direction over time. The comparison of two old laboratory-adapted strains, Canton-S and Harwich, suggested that they may be used as a multicellular eukaryote model for studying the effects of individual difference in aging processes on the physiological responses to various environmental factors, including low and high temperature, carbohydrate-heavy food, and caffeinated food in the presence of light-dark cyclicity and constant darkness. We found that the accelerated aging in Harwich was associated with a shorter lifespan, lower fecundity, faster rate of development, smaller body weight, lower concentrations of two main insect sugars, reduced locomotor activity, and larger amount of sleep. Such findings imply that, in the experimental studies of the aging accelerated effects of various adverse factors, the results obtained with only one of the laboratory-adapted strains cannot be generalized to other strains. The documented individual and strain-associated differences in the responses of their daily rhythms to aging and age-accelerated factors allow the recommendation to use at least two distinct strains, characterized by a range of relatively fast and relatively slow aging processes, for the experimental elaboration of relationships between genes, environment, behavior, physiology, and health.

## Figures and Tables

**Figure 1 insects-15-00329-f001:**
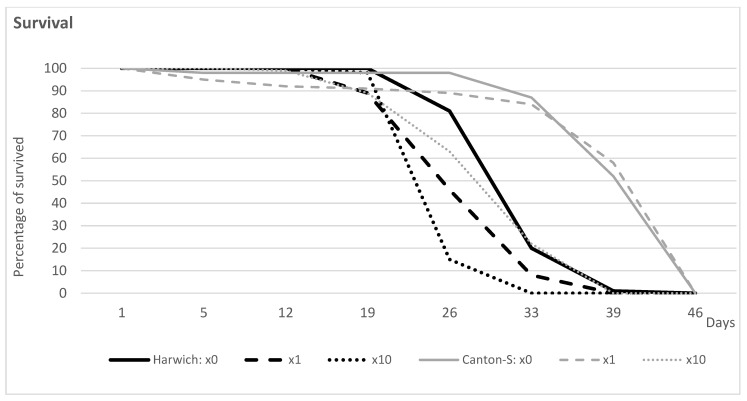
Survival curves obtained for males of the two strains kept at 29 °C and fed by standard food without (×0) and with different doses of caffeine-benzoate sodium (×1 and ×10).

**Figure 2 insects-15-00329-f002:**
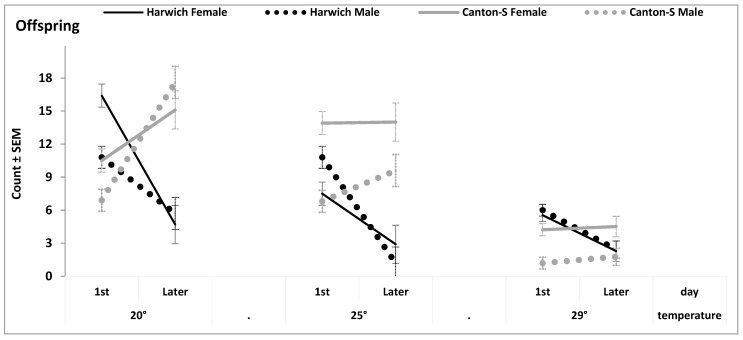
Fecundity under three different temperatures measured as the number of offspring per tube. Egg laying lasted for one day. Offspring were counted in the first and second (last) day of hatching. From the results of four-way rANOVA reported in Appendix A.

**Figure 3 insects-15-00329-f003:**
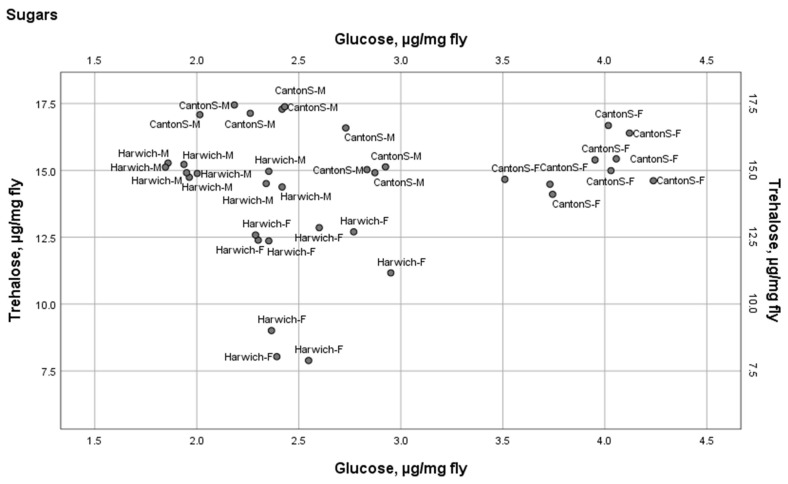
Two main insect sugars in male and female flies from the two strains. Harwich-M and Harwich-F and CantonS-M and CantonS-F: Harwich male and female and Canton-S male and female. From the results of three-way rANOVA reported in Appendix A.

**Figure 4 insects-15-00329-f004:**
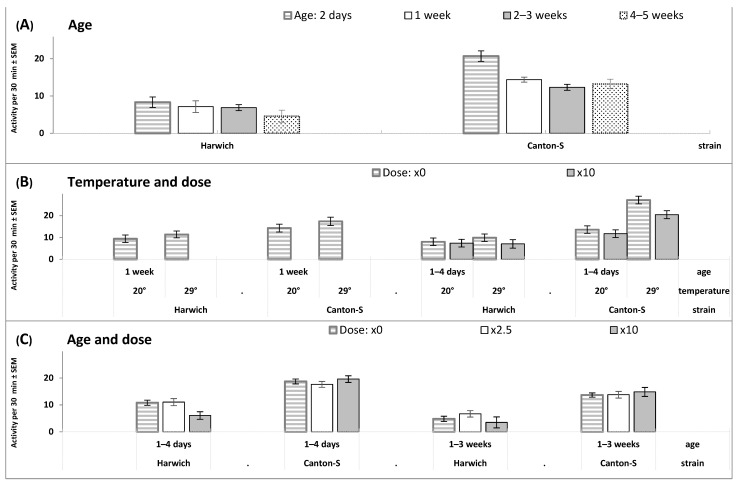
(**A**–**E**) Effects on daily averaged locomotor activity of aging, low and high temperature, carbohydrate and standard food, either without (×0) or with different doses of caffeine-benzoate sodium (×2.5 and ×10), constant darkness (DD), and 24 h light-dark cycle (LD). This figure is based on the results of the 6 rANOVA reported in Appendix A.

**Figure 5 insects-15-00329-f005:**
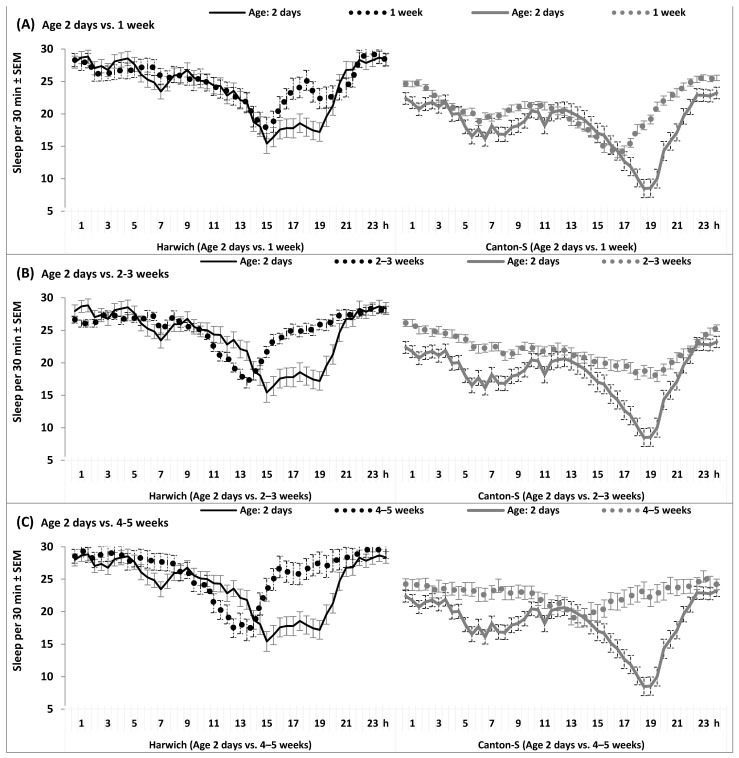
(**A**–**C**) The 24 h sleep pattern under high temperature in flies of four ages. This and remaining figures are based on the results of one of rANOVAs reported in Appendix A.

**Figure 6 insects-15-00329-f006:**
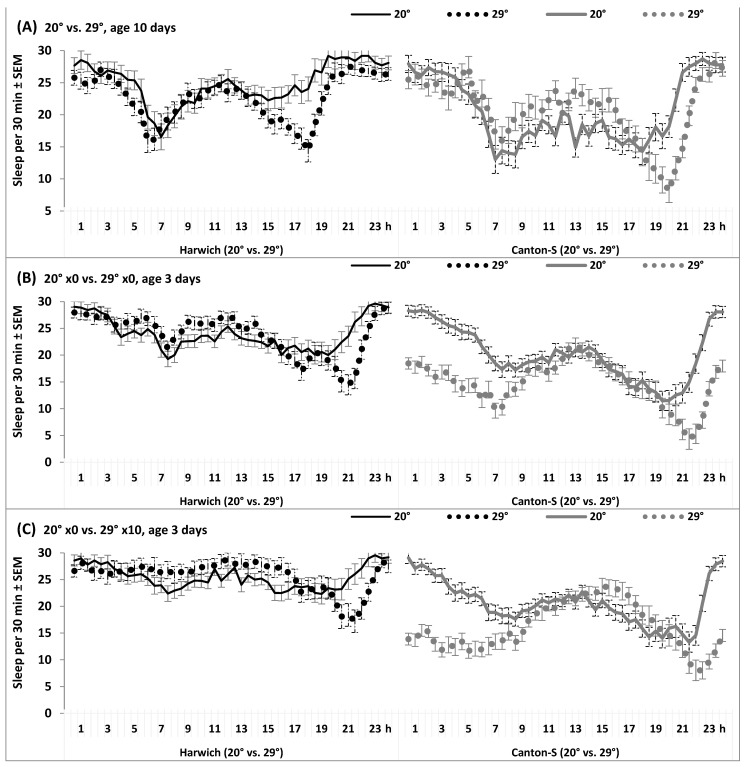
The 24 h sleep pattern under low and high temperature (**A**), and under low and high temperature without, ×0, or with 10 doses of caffeine-benzoate sodium solution, ×10 (**B**,**C**). (**A**) Age 10 days. (**B**,**C**) Age 3 days.

**Figure 7 insects-15-00329-f007:**
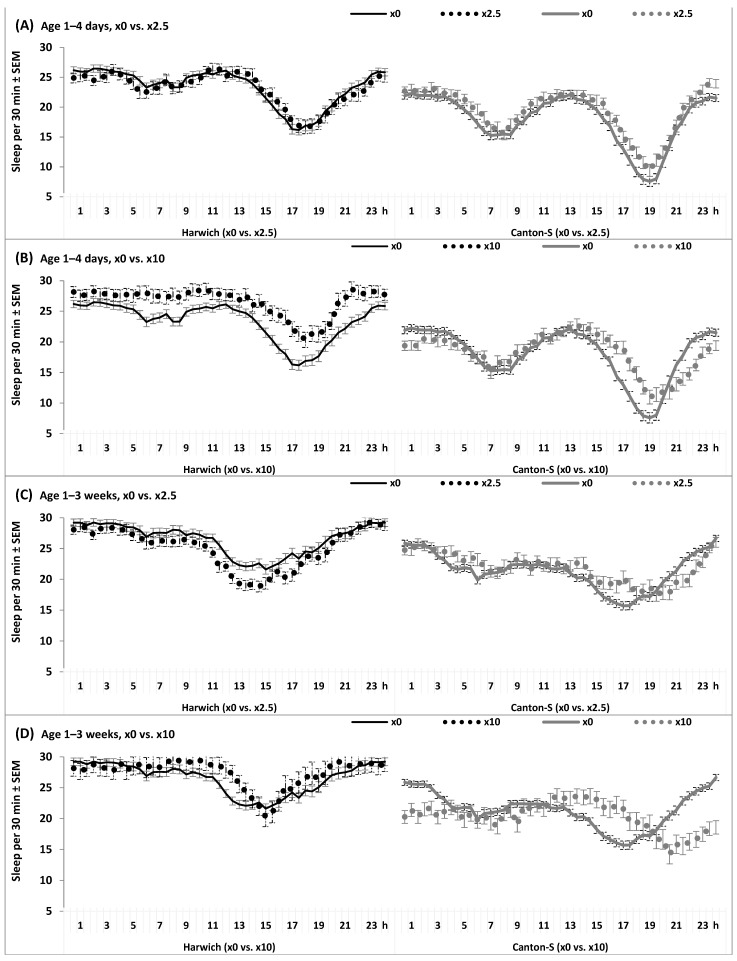
(**A**–**D**) The 24 h sleep pattern under high temperature at two ages and having food either without, ×0, or with 10 doses of caffeine-benzoate sodium, ×10. See also Figure 6.

**Figure 8 insects-15-00329-f008:**
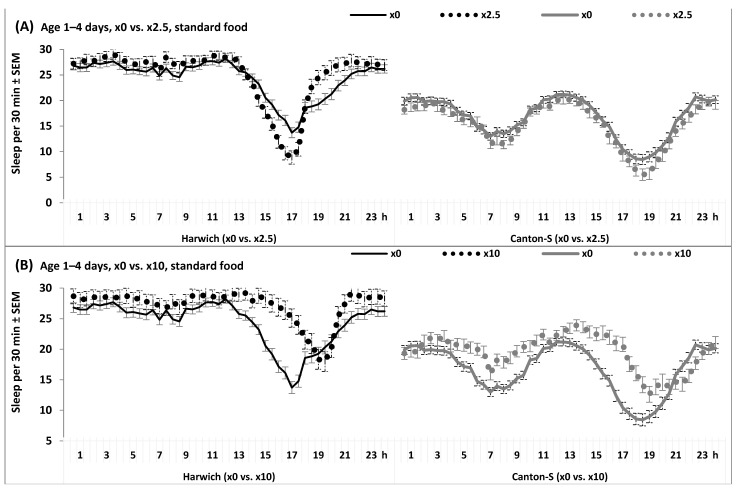
(**A**–**D**) The 24 h sleep pattern under high temperature, with carbohydrate or standard food without (×0) or with caffeine-benzoate sodium solution (doses ×2.5 and ×10).

**Figure 9 insects-15-00329-f009:**
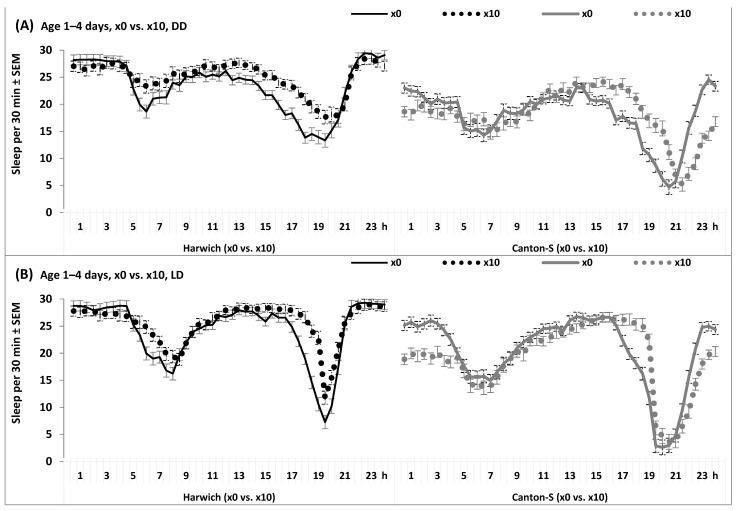
The 24 h sleep pattern in the condition of either constant darkness or under 24 h light-dark cycle. The effects of high temperature combined with standard food without (×0, (**A**)) or with 10 doses of caffeine-benzoate sodium (×10, (**B**)) See also legends to the previous Figure 4, Figure 5, Figure 6 and Figure 7.

**Figure 10 insects-15-00329-f010:**
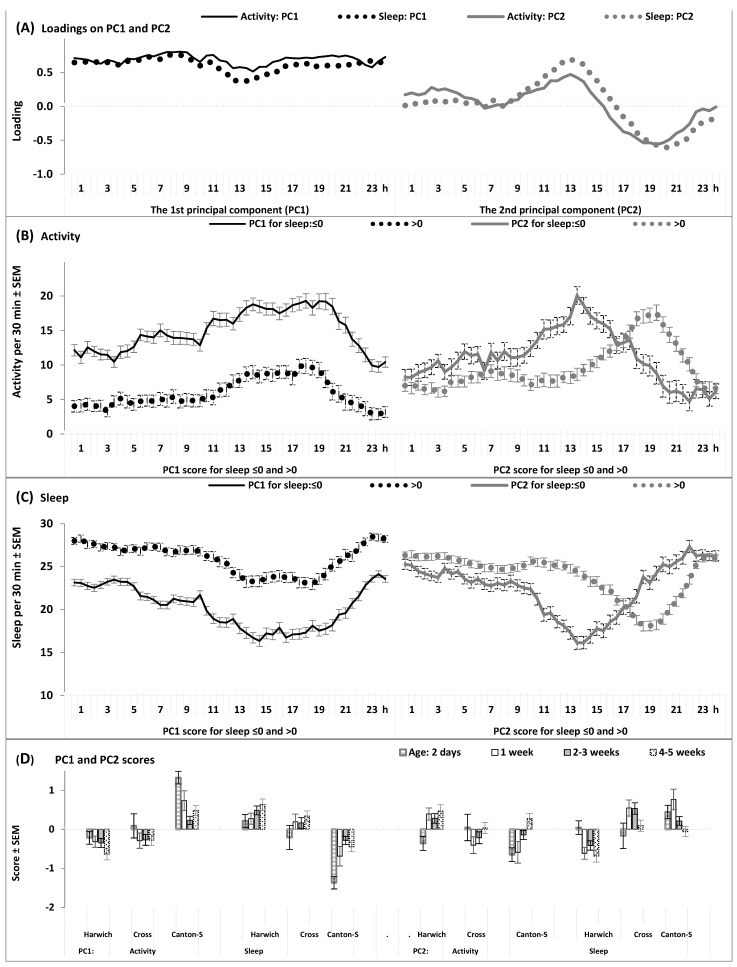
The 1st and 2nd principal components of the 24 h patterns of activity and sleep. (**A**) Loadings of 48 30 min intervals on the 1st and 2nd principal components (PC1 and PC2, respectively). (**B**,**C**) The 24 h patterns of locomotor activity (**B**) and sleep (**C**) of flies with PC1 and PC2 scores ≤ 0 and >0. (**D**) Rates of PC1 and PC2 scores ≤0 and >0 among flies of four ages from the parent strains and their crosses. See the results of statistical comparison of these rates in Appendix A.

## Data Availability

All data are available from the first author on reasonable request.

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
