# Peer review of "Two Old Wild-Type Strains of Drosophila melanogaster Can Serve as an Animal Model of Faster and Slower Aging Processes"

_insects, 2024, doi:10.3390/insects15050329_

Round 1
Reviewer 1 Report
Comments and Suggestions for Authors
An interesting study is presented here. The results appear to be sound. One concern was the lack of a methods section. Despite the methods being published elsewhere, a brief description is required.
Another concern was the lack of clarity in the conclusion. Lab strains used will vary in their genetics and will have different lifespans etc. For example, another common strain OreR lives quite long. Are the findings then not a great resource to make hybrid strains to isolate genes involved in longevity?
Comments on the Quality of English Language
Minor revision is required for the language. For example, the english word is dose not "doze" which appears throughout the manuscript.
Author Response
Reply to Review #1.
Top of Form
Bottom of Form
Top of Form
Top of Form
Bottom of Form
Top of Form
Review Report Form
Open Review
(x) I would not like to sign my review report
( ) I would like to sign my review report
Quality of English Language
( ) I am not qualified to assess the quality of English in this paper
( ) English very difficult to understand/incomprehensible
(x) Extensive editing of English language required
( ) Moderate editing of English language required
( ) Minor editing of English language required
( ) English language fine. No issues detected
|
Yes |
Can be improved |
Must be improved |
Not applicable |
|
|
Does the introduction provide sufficient background and include all relevant references? |
(x) |
( ) |
( ) |
( ) |
|
Are all the cited references relevant to the research? |
(x) |
( ) |
( ) |
( ) |
|
Is the research design appropriate? |
(x) |
( ) |
( ) |
( ) |
|
Are the methods adequately described? |
( ) |
( ) |
(x) |
( ) |
|
Are the results clearly presented? |
( ) |
(x) |
( ) |
( ) |
|
Are the conclusions supported by the results? |
( ) |
(x) |
( ) |
( ) |
Comments and Suggestions for Authors
An interesting study is presented here. The results appear to be sound. One concern was the lack of a methods section. Despite the methods being published elsewhere, a brief description is required.
Reply. The methods section was moved from Supplementary Material in the manuscript and extended.
Another concern was the lack of clarity in the conclusion. Lab strains used will vary in their genetics and will have different lifespans etc. For example, another common strain OreR lives quite long. Are the findings then not a great resource to make hybrid strains to isolate genes involved in longevity?
Reply. The conclusion section was extended to stress the main message of the prospect paper: the necessity to use several strains for accounting individual difference in the aging process and differential responses to adverse factors. The introduction section was extended to explain that the pair of strains, H and C-S, are of special interest for comparison because of their reproductive isolation under certain harsh environmental condition. Some results on the hybrids of these two strains were added to show involvement of many genes in longevity and the 24-h patterns of activity and sleep. A new paragraph about isolation of such genes in future studies was added in the discussion section.
Comments on the Quality of English Language
Minor revision is required for the language. For example, the english word is dose not "doze" which appears throughout the manuscript.
Reply. Thank you for this and other suggestion to improve the manuscript, such typos were checked and excluded, an English native speaker revised the language.
Submission Date
20 March 2024
Date of this review
04 Apr 2024 07:13:13

Reviewer 2 Report
Comments and Suggestions for Authors
I find this paper interesting and well-designed. Unfortunately, I have some suggestions for improvement of the paper. I summarize them below:
Reactive Oxygen Species Measurement
As reactive oxygen species seem to be one of the major causes of aging, it seems natural to measure ROS levels in the study strains and include this data in the paper. I believe that results showing how different strains handle ROS would improve the paper.
some references for that:
Lushchak, O.V. (2014) 'Free radicals, reactive oxygen species, oxidative stress and its classification', Chemico-Biological Interactions, 224, pp. 164–175. https://doi.org/10.1016/j.cbi.2014.10.016
Muñoz-Soriano, V. and Paricio, N. (2011) 'Drosophila as a model system to study na/k-ATPase function', Genes, 2(1), pp. 44–61. https://doi.org/10.3390/genes201004
Pomatto, L.C.D. and Davies, K.J.A. (2018) 'The role of declining oxidative metabolism in aging', F1000Research, 7, p. 171. https://doi.org/10.12688/f1000research.12510.
Materials and Methods Section
The lack of detailed materials and methods in the main manuscript makes it difficult to follow the experiment details. Unless the journal guidelines explicitly restrict including this section in the main text, it would be easier for readers if the materials and methods were incorporated into the primary manuscript rather than separated into supplementary materials.
Choice of Experimental Parameters
It is unclear why the authors chose to measure locomotor activity, sleep, high temperature, carbohydrate diet, and varying doses of caffeine-benzoate sodium as their experimental parameters. Some clarification on the rationale behind selecting these specific factors would help readers better understand the study design and hypotheses being tested.
In my opinion, providing more context around the choice of measurements and including detailed methods would significantly improve the manuscript's clarity and accessibility for readers. Unless prohibited by the journal, integrating these elements into the main text could enhance the overall flow and comprehensibility of the paper.
I congratulate the authors on their research idea and execution. They contribute very interesting data to the community studying fruit flies as well as those working on aging. I am confident that after incorporating the suggested revisions, your work will make a significant contribution to the scientific world.
Author Response
Reply to Review #2.
Review Report Form
Open Review
( ) I would not like to sign my review report
(x) I would like to sign my review report
Quality of English Language
( ) I am not qualified to assess the quality of English in this paper
( ) English very difficult to understand/incomprehensible
( ) Extensive editing of English language required
( ) Moderate editing of English language required
( ) Minor editing of English language required
(x) English language fine. No issues detected
|
Yes |
Can be improved |
Must be improved |
Not applicable |
|
|
Does the introduction provide sufficient background and include all relevant references? |
( ) |
( ) |
(x) |
( ) |
|
Are all the cited references relevant to the research? |
(x) |
( ) |
( ) |
( ) |
|
Is the research design appropriate? |
(x) |
( ) |
( ) |
( ) |
|
Are the methods adequately described? |
( ) |
( ) |
(x) |
( ) |
|
Are the results clearly presented? |
(x) |
( ) |
( ) |
( ) |
|
Are the conclusions supported by the results? |
(x) |
( ) |
( ) |
( ) |
Comments and Suggestions for Authors
I find this paper interesting and well-designed. Unfortunately, I have some suggestions for improvement of the paper. I summarize them below:
Reactive Oxygen Species Measurement
As reactive oxygen species seem to be one of the major causes of aging, it seems natural to measure ROS levels in the study strains and include this data in the paper. I believe that results showing how different strains handle ROS would improve the paper.
some references for that:
Lushchak, O.V. (2014) 'Free radicals, reactive oxygen species, oxidative stress and its classification', Chemico-Biological Interactions, 224, pp. 164–175. https://doi.org/10.1016/j.cbi.2014.10.016
Muñoz-Soriano, V. and Paricio, N. (2011) 'Drosophila as a model system to study na/k-ATPase function', Genes, 2(1), pp. 44–61. https://doi.org/10.3390/genes201004
Pomatto, L.C.D. and Davies, K.J.A. (2018) 'The role of declining oxidative metabolism in aging', F1000Research, 7, p. 171. https://doi.org/10.12688/f1000research.12510.
Reply. We added a special paragraph in Discussion about the prospects of future testing reactive oxygen species with referring to the publications of these 5 authors. We also added a sentence about our previous experience with testing reactive oxygen species in fertile and infertile female hybrids in one of our publications. Unfortunately, we cannot conduct a new experiment with testing reactive oxygen species for the present paper. First, it will take much more time than we have for this revision (the letter says: “Please revise your manuscript according to the referees’ comments and upload the revised file within 5 days”). Second, if the result of such an experiment would be included in the present paper, we must explain to the publisher why we added one additional author (who is the best expert on this methodology in our institute). Third, we think that it is wiser to prepare a new paper that can report the results of our deeper search for possible mechanisms of rapid aging process in one of the strains including the results of testing reactive oxygen species in two strains. Fourth, since, after inclusion Methods in the body of manuscript, as requested by both reviewers, we already exceeded the word limit of this journal, there is no room for any additional section.
Materials and Methods Section
The lack of detailed materials and methods in the main manuscript makes it difficult to follow the experiment details. Unless the journal guidelines explicitly restrict including this section in the main text, it would be easier for readers if the materials and methods were incorporated into the primary manuscript rather than separated into supplementary materials.
Reply. The methods section was moved from Supplementary Material in the manuscript and extended.
Choice of Experimental Parameters
It is unclear why the authors chose to measure locomotor activity, sleep, high temperature, carbohydrate diet, and varying doses of caffeine-benzoate sodium as their experimental parameters. Some clarification on the rationale behind selecting these specific factors would help readers better understand the study design and hypotheses being tested.
Reply. This chose of locomotor activity and sleep was explained by adding one paragraph to Introduction describing the list of tested life-history traits. This chose of adverse factors was explained by adding one sentence in the discursion section describing the expected bidirectional responses of the level of locomotor activity to these factors.
In my opinion, providing more context around the choice of measurements and including detailed methods would significantly improve the manuscript's clarity and accessibility for readers. Unless prohibited by the journal, integrating these elements into the main text could enhance the overall flow and comprehensibility of the paper.
Reply. We did this as recommended.
I congratulate the authors on their research idea and execution. They contribute very interesting data to the community studying fruit flies as well as those working on aging. I am confident that after incorporating the suggested revisions, your work will make a significant contribution to the scientific world.
Reply. Thank you very much, indeed.
Submission Date
20 March 2024
Date of this review
09 Apr 2024 15:17:41
